# Hole Matrix Mapping Model for Partitioned Sitting Surface Based on Human Body Pressure Distribution Matrix

**DOI:** 10.3390/healthcare11060895

**Published:** 2023-03-20

**Authors:** Chunfu Lu, Zeyi Chen, Yu Li, Xiaoyun Fu, Yuxiao Tang

**Affiliations:** School of Design and Architecture, Zhejiang University of Technology, Hangzhou 310023, China

**Keywords:** pressure distribution, seat cushion, ergonomics, comfort

## Abstract

(1) Objective: The objective of this study was to experimentally obtain the ideal pressure distribution model of buttock and thigh support for office workers in forward-leaning and upright sitting postures, reproduce the support provided by mesh materials with elastic materials, and propose an effective seat design scheme to improve the comfort of office workers. (2) Method: Based on the seven most popular mesh chairs on the market, pressure distribution experiments, and the fuzzy clustering algorithm, the relatively ideal body pressure distribution matrices were generated for office workers under two common sitting postures, and the corresponding partitioned sitting support surfaces were obtained. A prototype chair was created and validated by combining the ergonomics node coordinates and the physical properties of the materials. (3) Result: An ideal support model of four zones was constructed, and prototype pads were designed and produced according to this model. Subjects were recruited to test the ability of the prototypes to reproduce the ideal pressure distribution maps. (4) Conclusion: The four-zone ideal support model is capable of effectively representing the buttock and thigh support requirements in forward-leaning and upright sitting postures, and it is useful for the development of related products. Studying sitting postures and pressure values generated by different activities of office workers will help to refine the needs of office personnel and provide new ideas for the design of office chairs.

## 1. Introduction

In urban environments, heavy workloads force office workers to work at their desks continuously for long periods of time [1]. Over time, this will lead to muscle soreness in the waist, buttocks, and back, and even the deformation of the entire spine’s physiological curvature [2,3,4,5]. Sitting for a long time has become increasingly recognized as the culprit behind chronic diseases for many office workers [6]. The physical damage caused by sitting for a long time has become a major public health risk. A good seat surface can provide reasonable support for the buttocks and thighs, helping ensure good spinal curvature in office sitting postures and reducing the harm to the human body caused by sitting for a long time. Therefore, research on the comfort of the seating surface is of great importance.

There are more related studies for seat comfort [1,7,8,9,10], which provide significant theoretical bases for seat designs. The literature suggests that the support conditions for various contact areas between the seating surface and a human body should be varied for optimal comfort [11,12,13]. Vink and Lips found that the sensitivities of the buttocks and thighs are not the same, as the buttocks are less sensitive than the front of the thigh and the middle of the thigh, and the front of the thigh is more sensitive than the buttocks and the middle of the thigh [13]. Therefore, the support conditions of various parts and regions where the seat cushion and the human body come into contact should be different to optimize comfort.

In addition, the study of different sitting type states and their body pressure distribution is important for the study of seat comfort. For the definition of sitting posture, Chaffin [14] and Kilbom [15] point out that there is a positive relationship between musculoskeletal disorders and neck flexion over 20°. Drury [16] and Grandjean [17] define that a user is considered to be in a poor sitting posture if he or she is in a forward-leaning posture with trunk flexion over 20°. The study by KO’Sullivan [18] points out that the Subjectively Perceived Ideal Posture(SPIP) is only slightly different from the Tester Perceived Neutral Posture(TPNP), while the TPNP can be considered the real ideal sitting posture from a medical point of view. In terms of office sitting posture there are two main types of office sitting posture: forward-leaning posture and upright posture, with forward-leaning posture being the main working posture [19,20]. The forward-leaning sitting posture is mainly for operation and is a skillful posture; the upright sitting posture is mainly present in scenarios such as receiving/telephone calls. The forward-leaning sitting position was maintained for 11.6 min, and the upright sitting position was maintained for only 2.8 min [21], after which the trunk would alternate between forward and backward leaning [22].

Biomechanics is an essential consideration for office chair design. The results of a study by Cardoso [23] et al. suggest that split office chair design has the potential to bring physiological benefits to office workers. For the existing office chairs on the market, previous studies concluded that mesh chairs are more ergonomic compared to other types of chairs [24,25]. Guozhen Lu, Xianqing Xiong et al. [26] were to prove the effective support of the mesh material on the human body by analyzing the support relationship between the office population—chairs of five brands in the market (Ergomax, Herman Miller, Okamura, Dapoli, and Polyou). Li Lijuan et al. [27] pointed out that warp-knitted spacer fabric has excellent compression resistance and fastness to use, as well as good extensibility and flexibility, making it the best choice for office seating fabric and more comfortable to sit on. However, it has a series of problems, such as high-quality control requirements, complex processes, and the high cost of mesh fabric, which makes its selling price much higher than that of office chairs using memory foam as the seating surface material. The purpose of this study is to analyze the support model of the man–machine mesh chair and the possibility of reproducing it in an elastic material (memory foam) through pressure distribution and material studies.

Body pressure distribution is one of the very effective experimental methods and comfort evaluation methods to study the comfort of seating [28]. In addition to objective experiments, the most commonly used research methods are subjective evaluation methods [29,30,31,32,33]. A subjective evaluation method is based on the subjective evaluation results of subjects assessing seat comfort. This type of method has the advantage that it is relatively easy to quantify and analyze the feedback of respondents, but it has a long cycle, poor repeatability, and is prone to the psychological and physiological states of the subjects [34]. Objective experiments have higher repeatability, shorter cycle, lower study cost, and can objectively reflect seat comfort. In addition, objective experiments can also obtain data such as pressure distributions of sitting surfaces and stress of human soft tissues [19,35,36,37,38,39,40].

Studies have shown that compared with other types of chairs, ergonomic mesh chairs provide a good solution to physiological damages caused by sitting. However, issues such as stringent quality control requirements, complex manufacturing processes, and high cost limit their popularization and development. This study aims to analyze the possibility of reproducing the support model of the ergonomic mesh chair with an elastic material (specifically, memory foam) through pressure distribution and material studies.

This study was conducted in three stages. (1). First, pressure distributions on products available on the market were measured. In addition, combined with the subjective comfort evaluation of the subjects, the relatively ideal pressure distribution matrix was obtained through calculation. The zones were then determined by clustering. (2). Next, based on the zone support requirements, the hole array in the elastic support material (memory foam) was mapped out, and the prototype was made. The specific process is shown in Figure 1. (3). Finally, a back-testing experiment is conducted on the prototype model to observe the sitting comfort of different groups so as to verify the relevant knowledge extracted from the model.

## 2. Experiment on Human Body Pressure Distribution of the Buttocks and Thighs While Sitting

### 2.1. Experimental Subject and Method

#### 2.1.1. Experimental Apparatus and Parameter

In this experiment, the pressure distribution system made by Tactilus, Hangzhou City, Zhejiang Province, was used to collect the pressure distribution information of the subject’s sitting surface. The pressure sensor pad consisted of 32 rows and 32 columns, for a total of 1024 sensors. The pressure sensor software could calculate and return objective indicators such as maximum pressure, average pressure, and contact area. The pressure distribution parameters in the experimental analysis were the maximum pressure, the average pressure, the maximum pressure gradient, and the average pressure gradient.

The experimental apparatus used mesh chairs. After comparing 40 mesh chairs available on the market, we select 7 mesh chairs with the highest user ratings as the samples. All 7 mesh chairs could adjust sitting height, sitting depth, sitting surface inclination, lumbar height, lumbar angle, backrest depth, backrest height, backrest angle, and pivot point height and depth. They also allowed a transition between an upright sitting posture and a forward-leaning sitting posture. Their shapes and basic parameters are shown in Table 1.

To eliminate the interference of muscle activation, it was necessary to ensure that the subjects’ muscles were completely relaxed, so surface electromyography (sEMG) was used as an observable.

#### 2.1.2. Experimental Subjects

According to previous research [41], the main group of people using office chairs in China has the following characteristics: the overall level of education is relatively high, those involved in design are the class of people using office chairs more often, the main age group is 18–25 years old, and a higher percentage of women. This cohort made the ideal design target group.

A total of 7 healthy design graduate students, 4 males and 3 females, were recruited for the experiment. To reduce data variation caused by different body shapes, the subjects’ basic morphological parameters were as consistent as possible. The basic subject information is as follows: the male subjects were 24 ± 2 years old, 173.1 ± 3.3 cm tall, 70 ± 4.2 kg in weight, and the female subjects were 24 ± 2 years old, 162.1 ± 2.2. 5 cm tall, and 52.4 ± 3.1 kg in weight.

#### 2.1.3. Experimental Procedure

The test included two parts: a subjective evaluation test and an objective measurement.

As described in Section 1, the experiment took two sitting postures, forward-leaning and upright, and the sitting posture holding time was specified with reference to the findings of Sun Xinxin et al. [21]. The average time to maintain a forward-leaning sitting posture was 11.6 min, after which the torso would alternately lean forward and backward to relieve body fatigue, and the average holding time for upright sitting posture was only 2.8 min. Considering the limited holding time of the two sitting postures, in the experiment, subjects were required to maintain two sitting postures for 2 min for upright sitting and 10 min for forward-leaning sitting, respectively, while the experimental sitting posture criteria were determined as follows: upright sitting posture was determined as the user’s subjective perceived ideal sitting posture. The forward-leaning sitting posture was a forward tilt of the trunk centerline of less than 20° [42].

Before starting the test, subjects were allowed to familiarize themselves with the seat for at least 2 min. When collecting body pressure distribution data, subjects were allowed to take a 2-min break in the process of position switching to reduce fatigue. Between tests with different seat types, subjects were allowed to stand up and move slightly for 5 min.

During the test, to ensure that the subjects’ muscles were completely relaxed, EMG signals from the transverse erector spinae (located on both sides of the spine in the waist and in the pelvis) and the gluteus maximus (located on the posterolateral side of the pelvis) that are most closely related to the posture of the buttocks and thighs were monitored as observation data [43],. As shown in Table 2 and Figure 2. When the EMG showed electrical silence, the muscles were considered to be fully relaxed, so the pressure distribution data was valid [44].

At the end of the test, the subjects were asked to fill out an experience assessment form. The assessment was done with reference to six indicators mentioned in the paper: shoulder, back, lower back, hip, thigh and overall [11]. Each item was scored on a 5-point scale, with 1 being extremely uncomfortable and 5 being extremely comfortable, up to a maximum score of 30.

#### 2.1.4. Standardization

The results of the study by kazushige et al. [45] showed that the cushion area could be divided as shown in Figure 3, with zone A being the 4 cm × 4 cm area under the sciatic bone, zone B being the 10 cm × 10 cm area around zone A, zone C being the entire buttock area, and zone D being the area outside zone C and including the leg area, and based on the results of this study, the size of the sciatic tuberosity area was obtained. This was to facilitate the acquisition and processing of body pressure distribution data, to predict the position of the sciatic tuberosity, to ensure that the sciatic tuberosity position was basically the same when the subjects were in the experiment, and to ensure that the data was not shifted when the body pressure distribution matrix was subsequently averaged to ensure accuracy.

### 2.2. Statistical Analysis

The acquisition of the ideal pressure distribution matrix needed to be determined by calculating the body pressure distribution parameters. The meaning of each parameter is described as follows.

#### 2.2.1. Measurement Data Analysis

The contact area, maximum pressure, and average pressure were directly read out by BPMS. The ASCII matrix pressure distribution information saved during the experiment was imported into Excel. The average pressure gradient and maximum pressure gradient were calculated from the pressure distribution matrix.

#### 2.2.2. Analysis of Subjective Evaluation Results

The subjective evaluation method of the experiment adopted the semantic differentiation method [46,47]. The five-degree adverbs from small to large were numbered 1–5, of which 1 was the least comfortable, 5 was the most comfortable, and the highest score was 20 points. The subjects were asked to subjectively evaluate the corresponding experimental items with numbers during each experiment.

### 2.3. Results and Analysis

#### 2.3.1. Calculation of the Similarity of Body Pressure Distribution Maps

For matrix similarity, the cosine similarity equation below is used to calculate the similarity of the pressure distribution matrices.
(1)$$similarity=\frac{A\times B}{\left\|A\right\|\left\|B\right\|}=\frac{\sum_{i=1}^{n}A_{i}\times B_{i}}{\sqrt{\sum_{i=1}^{n}{\left(A_{i}\right)}^{2}}\times \sqrt{\sum_{i=1}^{n}{\left(B_{i}\right)}^{2}}}$$

Here Ai and Bi represent the values of pressure points of the ideal pressure distribution matrix and the measured prototype pressure distribution matrix, respectively. The closer the calculated similarity is to one, the more similar the two matrices are. The similarity of the body pressure distribution maps between males and between males and females was calculated separately, and the similarity values were averaged. The obtained results are shown in Table 3. The results show that the similarity of pressure distribution maps between the same sexes is greater than that between opposite sexes.

#### 2.3.2. Human Body Pressure Distribution Maps for Top 10% Comfort Ranking

In order to select the best among the best, the pressure distribution maps obtained from the experimental results were ranked according to the subjective evaluation of the subjects, and the top 10% of body pressure distribution maps were selected for comfort.

According to the results of 2.3.1, the pressure distribution maps were classified according to gender and divided into four categories: male upright and forward sitting, female upright and forward sitting. Figure 4 shows the human body pressure distribution of the top 10% of the samples in the comparative test.

#### 2.3.3. Human Body Pressure Distribution Indicators

The key body pressure distribution indicators are shown in Table 4. The seats and subjects with the highest subjective evaluation of comfort in the forward-leaning and upright sitting postures are E/S7, A/S5, D/S7, and F/S6 for males and C/S1, D/S3, E/S3, and F/S2 for females as.

## 3. Ideal Pressure Distribution Matrix and Zone Partition

### 3.1. Ideal Pressure Distribution of Sitting Posture

The top 10% body pressure distribution map of comfort in Section 2.3.2 was differentiated according to gender and averaged to obtain an approximate ideal pressure distribution map, as shown in Figure 5. In Darwin’s theory of evolution by natural selection, the averageness hypothesis states [48] that the average value of a feature is better than the extreme values. Then, to a certain extent, the pressure distribution matrix with a higher comfort level can represent the approximate ideal pressure distribution matrix.

### 3.2. Ideal Pressure Distribution Matrix and Its Zones

The above four approximate ideal pressure distribution matrices are averaged according to gender again to obtain an approximate ideal pressure distribution matrix and then carry out Fuzzy C-means Clustering (FCM) on this matrix, and the mathematical model is as follows, based on the obtained results, the ideal pressure distribution map can be partitioned intuitively and effectively.
(2)$$J_{m}\left(\mu ,V\right)=\sum_{i=1}^{n}\sum_{k=1}^{c}\mu_{ik}^{m}{x\left\|x_{i}-v_{k}\right\|}^{2}$$
where n represents the number of pressure points, c represents the number of zones, vk represents the cluster center of the k-th class, μik represents the degree of membership of the i-th sample belonging to the k-th class, ‖xi − vk‖2 represents the squared Euclidean distance from the sample xi to the cluster center vk, and m represents the fuzzy index, which is generally equal to two. The results of FCM clustering on the matrices are shown in Figure 6.

Combined with the actual situation of sitting ergonomics, the ideal pressure distribution was mapped to the size of each area on the seat, as shown in Figure 7. The body pressure distribution indexes of each sub-area of the above pressure distribution were extracted and calculated as shown in Table 5, of which Area A was a non-pressure area and is not included in the statistics.

## 4. Prototype Production of Ideal Pressure Distribution

### 4.1. Determination of Key Ergonomic Node Coordinates

The anatomical software 3D body was used to display the subcutaneous tissue of the buttocks and thighs. The zone partition is made according to the ideal support zone sizes, as shown in Figure 6. In this study, 10 buttock and thigh histology experts were invited to use Figure 8 as an object to compare and evaluate the partition of the seven zones for the two groups of subjects. Analysis was conducted based on the evaluation and comparison results. The comparison between zones and corresponding key ergonomic nodes is shown in Table 6.

According to the human dimensions of Chinese adults provided by the China national standard GB10000-88, 573 ergonomic data samples of human body dimensions in sitting posture were collected using a shape ruler and a Martin measuring instrument. The samples were classified according to gender, and the contour curves of the buttocks and thighs of different genders were obtained. The ergonomic nodes corresponding to each zone in the curve are marked, and the median of the coordinates of each node in the group was calculated. The results are shown in Figure 9.

### 4.2. Protype of Ideal Cushion

By mapping the corresponding ergonomic node positions of each zone obtained above to the seat model, the shape and size of the seat after compression can be obtained, as shown in Figure 10.

According to Equation (3), expressing the equivalent elastic coefficient in our previous study [49], the height and pressure distribution values of each area are used in the calculation to obtain the equivalent elastic coefficient required by each area.
(3)$$k_{i}=∆F_{i}/∆H_{i}$$
where k_i_ is the comprehensive stiffness coefficient of the material of the i-th zone, ΔFi is the total pressure of the i-th zone, and ΔHi is the height change.

According to the experimental method for studying the relationship between pore diameter and compression deformation [49], memory foam with an 8 cm thick cushion pad, a density of 35D, and a staggered hole spacing of 20 mm was selected. The pore diameter D is a variable. A rod with a 10 cm diameter disk end is used to simulate head pressure. The results are obtained by linear fitting. The relationship between the pore diameter and the equivalent elastic coefficient is shown in Table 7.

The pore diameter is used as an independent variable to fit linearly to the equivalent elastic coefficient k value, as shown in Figure 11. The exponential function that has good goodness of fit is shown below:(4)$$D=26.341e^{-3.058k}(8\leq D\leq 20)$$

The distribution of the mapped hole matrix is obtained using the RHINO parametric modeling tool, as shown in Figure 12a,b. According to these distributions, 35D polyester fiber is used as the material to make a standard ideal support pillow prototype, as shown in Figure 12c. The equivalent elastic coefficient values of the key ergonomic node mapping positions in each area are measured, as shown in Table 8.

## 5. Experimental Study on the Model of Ideal Cushion Prototype

### 5.1. Experiment on Pressure Distribution of the Model of Ideal Cushion Prototype

Five eligible subjects from each of the two groups participated in the verification experiment using the corresponding ideal cushion prototype models and following the procedure described in Section 2. The chair with the highest comprehensive score in the first experiment was selected for the control group. The subjects were asked to make subjective scores after the test, and the evaluation indicators included firmness, wrapping, support, and fit. Each item was rated on a five-point scale, with one being the least comfortable and five being the most comfortable, for a maximum possible score of 30. The resulting pressure profile is shown in Figure 13.

### 5.2. Results and Analysis

#### 5.2.1. Comparative Analysis of Data for Each Sex

The analysis method described in Section 2 is used for the comparative analysis of data on the two sitting postures. The average pressure, peak pressure, maximum pressure gradient, average pressure gradient, and ideal values of each zone are compared between the prototype and the control group (Figure 14). The following information can be found, as shown by the mean pressure, the data range of each region of the prototype is significantly narrower and more concentrated in distribution, and the median value of the prototype is closer to the ideal value described in Table 4 when compared with the control group. The values of subdivisions B and D are closer to the ideal value, with only +0.09 and +0.11 for males and females in subdivision B, respectively, and −0.01 and—The peak pressure structure is similar to the mean pressure, with the median values in each region of the prototype being closer to the ideal, especially in subpart B, which shows a better approximation with a difference of +0.46 and −0.07. In terms of the maximum pressure gradient, region B of the prototype has an advantage of +0.06 and +0.19 for males and females, respectively. On the contrary, in females, the values in regions C and D are significantly lower than the ideal values. In terms of mean pressure gradient, the median value distribution of each subdivision of the prototype was better than that of the control group, but the difference was not significant, with region C being the closest and almost equal to the ideal value. In summary, the data distribution of the prototype tends to be more concentrated. The data range of the prototype is significantly reduced, and the median values are closer to the ideal values. In particular, the median values of zones C and D are closer to the ideal value.

The body pressure data in each zone of the prototype are superior to the data of the control group, among which the peak pressure and maximum pressure gradient are superior. The data groups of maximum pressure gradient and average pressure gradient have fewer data scattering and tighter distributions. However, for the maximum pressure values, the prototype has a certain deviation from the control group. This may be because the hole size distribution disperses the pressure, avoiding pressure concentration. Thus, it can be judged that each body shape of the office population can be included because the hole array memory foam pad can better restore the pressure distribution map of the mesh support. The ideal pressure distribution map is not limited to a single group of people because it can cover all body shapes within the office population.

In addition to restoring the ideal body pressure indicators in each partition, the similarity of the pressure distribution matrix is also an important basis for evaluating whether the prototype has restored the ideal support surface. The prototype, reference sample, ordinary sponge cushion and ideal pressure distribution matrix were calculated separately for similarity, and the results are shown in Figure 15 below. Among them, the ordinary sponge cushion is the average perforated sponge cushion (hole specification is the smallest hole diameter of 4.2), and the size specifications are all consistent with the sponge used in the prototype. The results show that the similarity between each sample and the ideal pressure distribution map is much higher than that of the ordinary sponge cushion, and the performance is also superior compared with the reference sample. The similarity of the prototypes to the ideal pressure distribution matrix was relatively consistent across all subjects in the male data, indicating that the ideal support for men was achieved. In contrast, the female data showed greater fluctuations, with no significant difference in the similarity of the prototypes compared to the reference sample. Only some (3/10) of the samples had better similarity than that of the reference sample, with the remaining others (7/20) having slightly lower similarity than the control pillows. The support surface requirements for women are more complex compared to men and require more experimental studies.

Uniform pressure distribution can characterize better comfort, according to Ahmadian [50], where they characterized the ability of the cushion to produce uniform pressure distribution. The seat pressure distribution index (Seat Pressure Distribution SPD%) was proposed and calculated as follows:(5)$$SPD\%=\frac{\sum_{i=1}^{n}{\left(p_{i}-p_{m}\right)}^{2}}{4np_{m}^{2}}\times 100$$
where Pi is the pressure on the i-th unit; Pm is the average pressure; n is the total number of pressure points with non-zero values; SPD% can be used for the calculation of pressure distribution in static and dynamic environments, SPD% characterizes the uniformity of the overall body pressure distribution on the cushion, the smaller the SPD%, the more uniform and the higher the overall comfort evaluation of the seat. The results are shown in Table 9 below. It can be seen that the SPD% value of the sample in upright sitting posture is lower than that of the reference sample, the values of the reference sample in forward sitting posture are higher than those of the reference sample, and the SPD% values of the male sample are similar to those of the reference sample, while the female data fluctuate relatively more. In the upright sitting posture, the sample shape surface and human buttocks and thighs fit better, SPD% is relatively small, the overall distribution of pressure is more uniform, to a certain extent, has been able to restore the overall comfort of the sample; forward sitting posture, male and female sample fit compared to the reference sample fit and a certain gap. Compared with the upright sitting posture, the forward sitting posture is more complex, and more factors need to be considered in the design.

#### 5.2.2. Subjective Scoring Analysis

The overall subjective score shows a downward trend with the experiment time. The score of each body part is calculated by Equation (6):(6)$$C=1-\overset{-}{N}/N_{max}$$
where N is the mean of each subjective evaluation item, and Nmax is the maximum value of subjective evaluation. The larger the C value, the higher the subjects are evaluated. According to the equation, the total comfort value of the subjective evaluation obtained by the above questionnaire is calculated, as shown in Table 10.

The group with the best subjective evaluation in the first human body pressure test experiment is selected as the reference group. Calculation shows that the subjective evaluation values of the prototype model are closer to the control group. The subjective score in the upright sitting position is higher than that in the control group. However, during working hours, the duration of the forward-leaning sitting posture is longer than that of the upright sitting posture, so the four major partitions of the seat surface for forward-leaning may not be enough; further partitions may be required.

The prototypes are sorted from low to high according to the similarity of the ideal pressure distribution matrices, and the subjective evaluation scores of the prototypes obtained are compared, as shown in Figure 16. The trend between subjective evaluation and similarity is somewhat consistent. The agreement between the subjective measurements and similarity was slightly greater for men than for women. It can be seen that the similarity with the ideal pressure distribution matrix characterizes the comfort evaluation to a large extent and also reflects that the ability to reproduce the ideal pressure distribution matrix is an important indicator for evaluating the comfort of the sitting surface.

## 6. Conclusions

The main conclusions of this study are as follows:(1)Through the analysis of the ideal pressure distribution map, the sitting contact surface can be divided into three zones: the ischial zone, the buttock zone, and the thigh zone. The ideal body pressure distribution index of each zone is shown in Table 4.(2)The similarity of the pressure distribution of a seat cushion to the ideal pressure distribution reflects comfort to some extent. The similarity of the pressure distribution matrices is compared with the subjective comfort scores. The two are highly consistent. Whether or not the ideal pressure distribution matrix can be reproduced more accurately is an important indicator for evaluating the comfort of the seat cushion.(3)The partitioned hole matrix mapping method can reproduce pressure distribution maps between different materials (e.g., mesh surface and memory foam). This method can be used to design memory foam chairs that perform competitively with existing mesh chairs and can be fabricated more easily for a lower cost. Therefore, this method is important for the design and development of ergonomic chairs.

## 7. Future Prospects

Through the experiment of the body pressure distribution of existing mesh chairs, the ideal pressure distribution map of office workers in the two most common sitting postures during working hours was studied. Based on this, the optimization scheme for the seat surface of elastic support materials is designed. However, we cannot rule out the possibility that we have missed a better support scheme, so the resultant ideal pressure distribution matrix can only be said to be relatively optimal to a certain extent. In addition, in the model prototype back-testing experiment, only gender is classified, and the differences in physique among populations have not been refined. Furthermore, in the course of the experiment, factors such as conversations with researchers could cause unnatural changes in posture, and the experimental posture is relatively simple and does not take into account the non-standard sitting postures of users. All these limitations should be addressed in future research.

## Figures and Tables

**Figure 1 healthcare-11-00895-f001:**
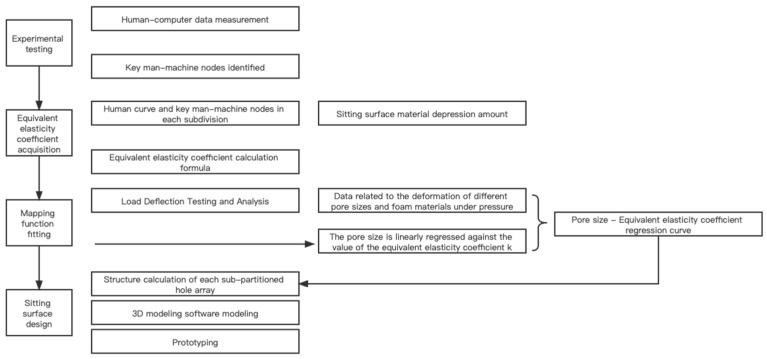
Technology Line.

**Figure 2 healthcare-11-00895-f002:**
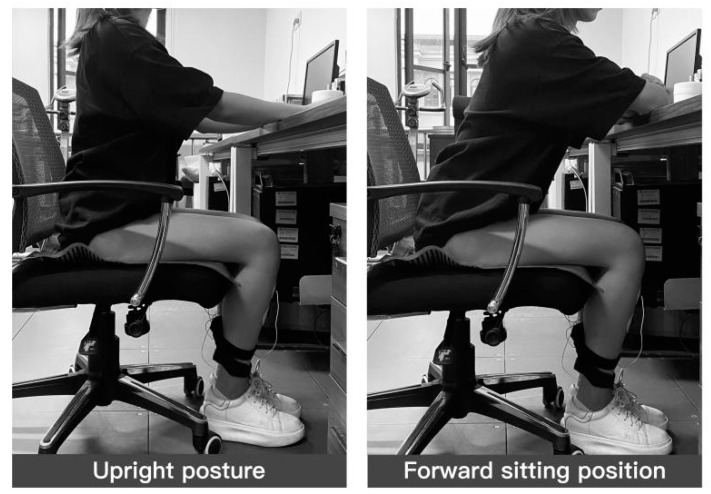
Pressure distribution data under sEMG monitoring.

**Figure 3 healthcare-11-00895-f003:**
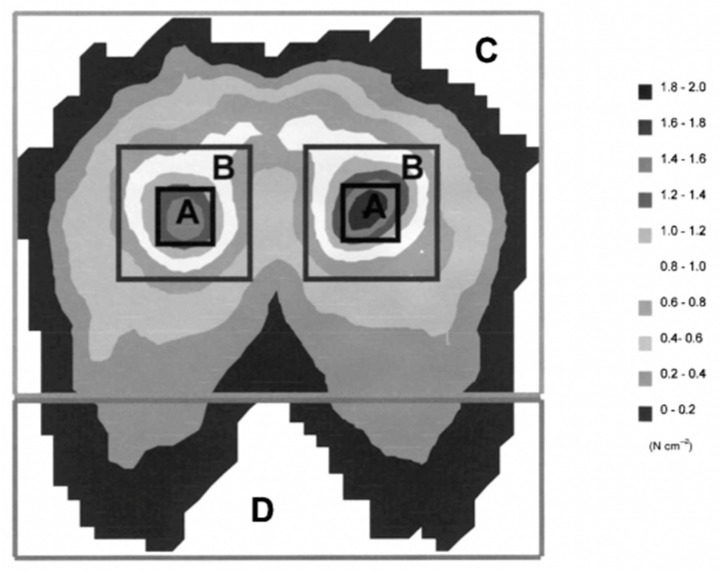
Cushion body pressure distribution area division.

**Figure 4 healthcare-11-00895-f004:**
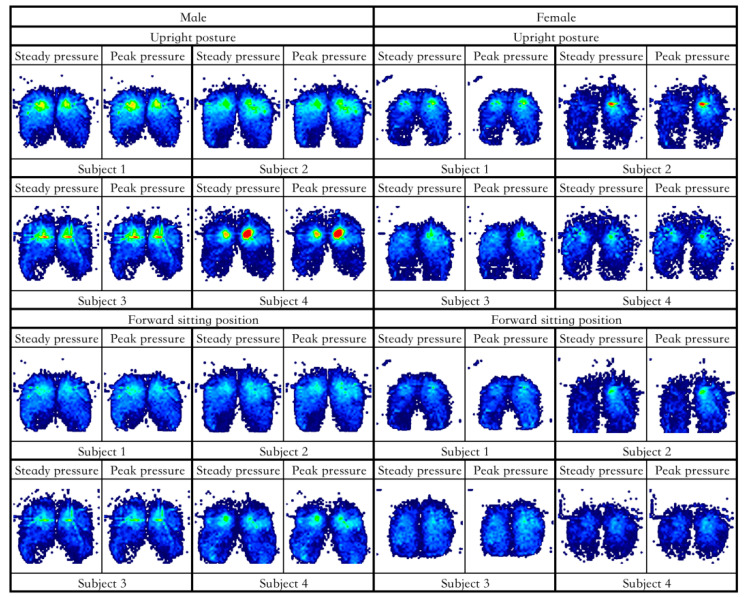
The cloud map of pressure distribution in upright and forward sitting posture with the highest comfort rating.

**Figure 5 healthcare-11-00895-f005:**
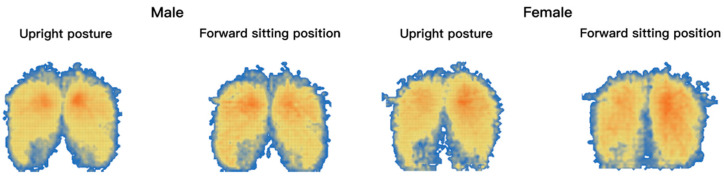
Approximate ideal pressure distribution.

**Figure 6 healthcare-11-00895-f006:**
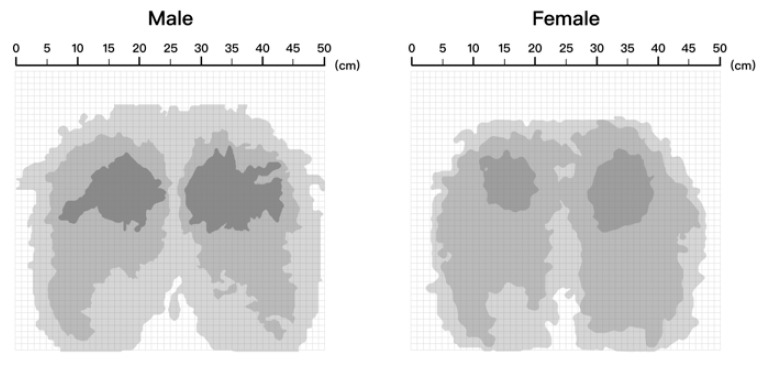
Approximate ideal pressure distribution matrix partition.

**Figure 7 healthcare-11-00895-f007:**
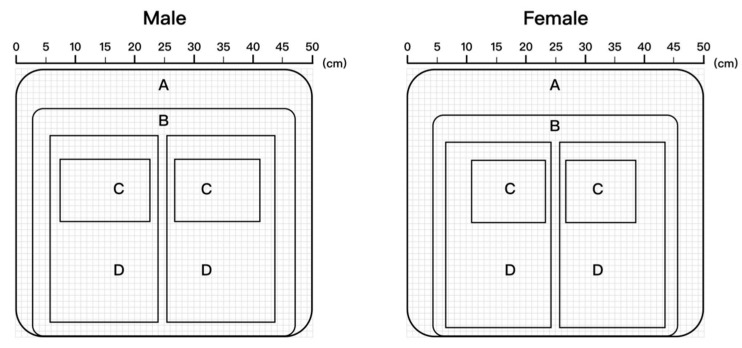
Seat surface partition.

**Figure 8 healthcare-11-00895-f008:**
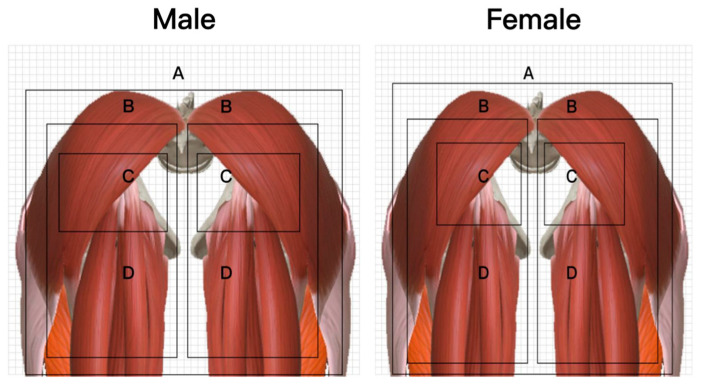
Subcutaneous tissue and structure of each subdivision.

**Figure 9 healthcare-11-00895-f009:**
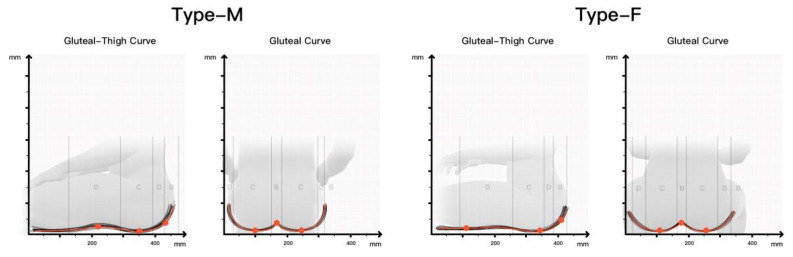
Human–machine node median coordinates of hip and leg profiles of 2 groups.

**Figure 10 healthcare-11-00895-f010:**
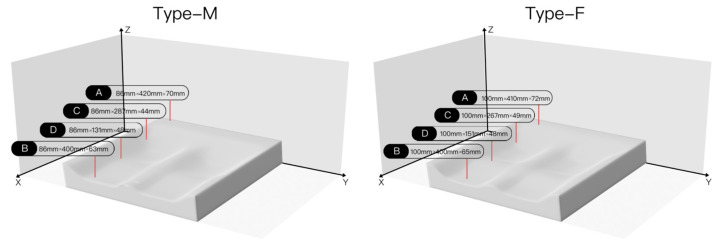
The two groups of people have the post-compression form.

**Figure 11 healthcare-11-00895-f011:**
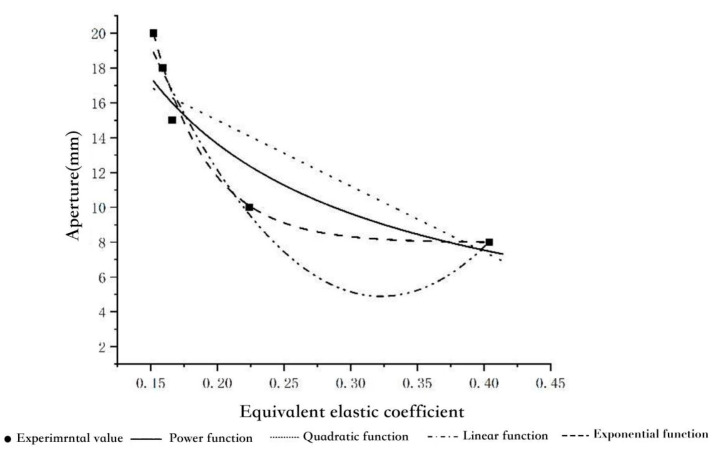
Pore size–equivalent elastic coefficient regression curve.

**Figure 12 healthcare-11-00895-f012:**
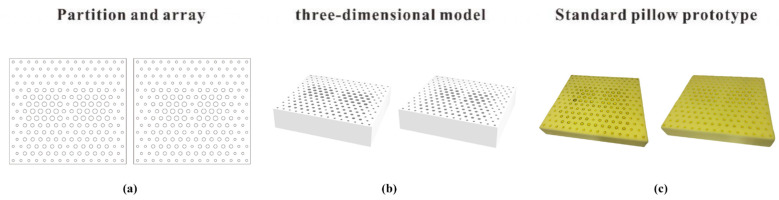
Hole array distribution & 3D model & Prototype.

**Figure 13 healthcare-11-00895-f013:**
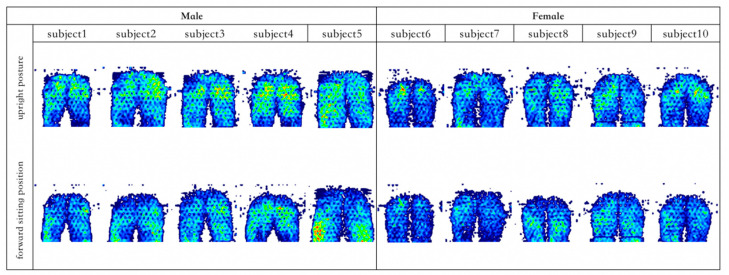
Prototype pressure distribution diagram.

**Figure 14 healthcare-11-00895-f014:**
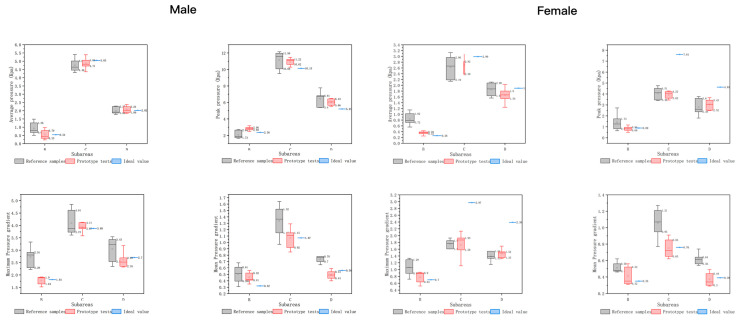
Comparison of body pressure distribution indexes in subregions of two groups of population.

**Figure 15 healthcare-11-00895-f015:**
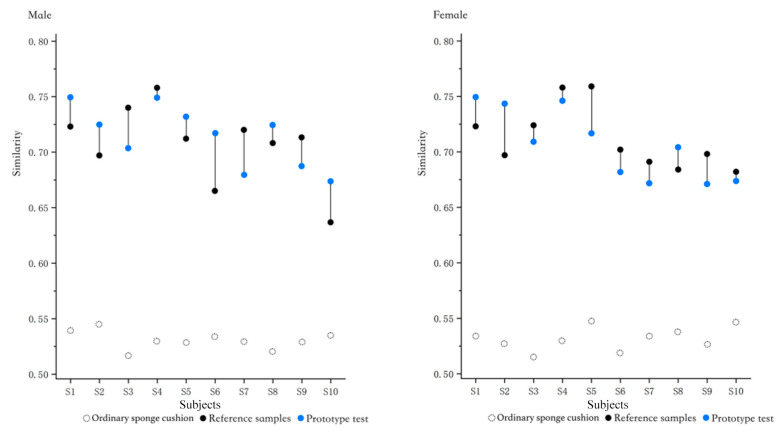
Comparison between the similarity of the prototype test, reference samples, ordinary sponge cushion and ideal pressure distribution.

**Figure 16 healthcare-11-00895-f016:**
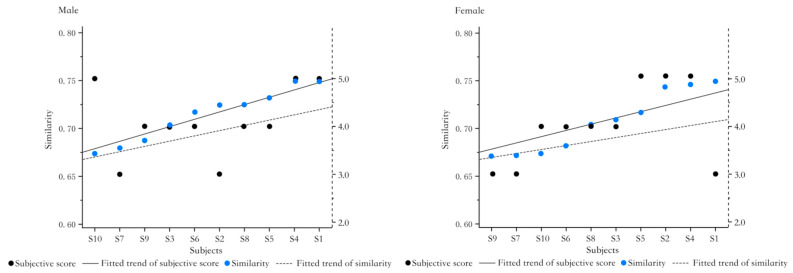
Ideal pressure distribution, subjective evaluation of prototype similarity.

**Table 1 healthcare-11-00895-t001:** 7 types of man–machine net chairs.

	A	B	C	D	E	F	G
	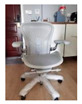	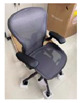	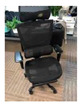	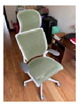	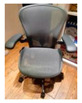	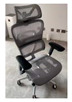	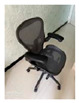
Length (mm)	500	500	600	600	500	600	500
Width (mm)	400	400	450	450	400	450	400
Height (mm)	60	50	60	60	50	50	70
material	Mesh fabric	Mesh fabric	Mesh fabric	Mesh fabric	Mesh fabric	Mesh fabric	Mesh fabric

**Table 2 healthcare-11-00895-t002:** Instructions for the Location of the Muscle to be Tested and the Electrode Sheet to be Pasted.

Name of the Target Muscle	Psoas	Gluteus Maximus
Electrode location	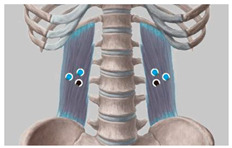	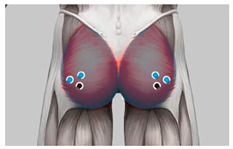

**Table 3 healthcare-11-00895-t003:** Volume pressure profile similarity calculation.

Male	Male and Female
0.832	0.581

**Table 4 healthcare-11-00895-t004:** Consistency test of evaluation matrix.

	Male							
	Upright Posture				Forward Sitting Position			
	E/S7	A/S5	D/S7	F/S6	E/S7	A/S5	D/S7	F/S6
Comfort score	27	25	24	24	27	25	24	24
Average pressure (Kpa)	3.50	3.46	3.25	2.86	3.18	2.91	2.66	2.81
Peak pressure (Kpa)	3.69	3.58	3.38	2.96	3.28	3.16	2.82	3.06
Maximum pressure gradient (Kpa/m^2^)	1.67	0.98	1.13	1.07	1.02	0.90	1.00	0.95
Mean pressure gradient (Kpa/m^2^)	1.08	0.95	1.15	1.01	1.01	0.85	0.97	0.88
	**Female**							
	**Upright Posture**				**Forward Sitting Position**			
	**C/S1**	**D/S3**	**E/S3**	**F/S2**	**C/S1**	**D/S3**	**E/S3**	**F/S2**
Comfort score	29	28	28	27	29	28	28	27
Average pressure (Kpa)	2.76	2.12	2.71	2.32	2.35	2.05	2.75	1.86
Peak pressure (Kpa)	3.08	2.26	2.88	2.41	2.83	2.27	3.08	2.07
Maximum pressure gradient (Kpa/m^2^)	1.05	0.95	0.90	1.10	1.01	0.85	0.95	0.79
Mean pressure gradient (Kpa/m^2^)	0.97	0.90	0.87	1.80	0.85	0.77	0.87	0.73

**Table 5 healthcare-11-00895-t005:** Sub-partition ideal volume pressure distribution index.

Male			
	B	C	D
Average pressure	0.54	5.05	2.02
Peak pressure	2.36	10.13	5.21
Maximum pressure gradient	1.81	3.88	2.70
Mean pressure gradient	0.32	1.07	0.56
**Female**			
	**B**	**C**	**D**
Average pressure	0.26	2.99	1.90
Peak pressure	0.89	7.61	4.63
Maximum pressure gradient	0.70	2.97	2.39
Mean pressure gradient	0.38	0.76	0.39

**Table 6 healthcare-11-00895-t006:** Each sub-partition corresponds to a man-machine node.

B	C	D
Sacral Triangle Diamond Unit	Buttock Units	Thigh Units
Femoral biceps	ischial tuberosity	hip joint

**Table 7 healthcare-11-00895-t007:** The measured equivalent elastic coefficient at different aperture (D).

	S1	S2	S3	S4	S5
D (mm)	8	10	15	18	20
K	0.414	0.214	0.166	0.159	0.152

**Table 8 healthcare-11-00895-t008:** Equivalent elastic coefficient of each subpartition.

	B	C	D
K (Calculated)	0.27	0.15	0.19
K (Actual)	0.25	0.14	0.21

**Table 9 healthcare-11-00895-t009:** Comparison of prototype test and reference sample SPD% values.

Male						
		S1	S2	S3	S4	S5
Upright posture	prototype test	10.49	13.02	11.18	12.80	13.74
	reference sample	12.20	14.69	11.42	14.25	16.21
Forward sitting position	prototype test	10.36	14.39	10.38	10.45	15.21
	reference sample	8.66	12.43	9.05	10.36	14.79
**Female**						
		**S1**	**S2**	**S3**	**S4**	**S5**
Upright posture	prototype test	11.73	12.05	13.69	9.56	12.57
	reference sample	11.80	15.49	16.43	10.99	16.97
Forward sitting position	prototype test	11.64	13.11	11.68	10.23	13.06
	reference sample	9.27	9.65	10.94	7.58	13.78

**Table 10 healthcare-11-00895-t010:** Subjective evaluation of total comfort.

		Hardness	Packing	Support	Fit	Subjective Feeling
Reference samples	Upright posture	1.012	1.020	0.842	1.152	0.526
	Forward sitting position	1.392	0.985	0.816	1.176	0.941
Prototype tests	Upright posture	1.125	1.188	0.875	1.021	0.648
	Forward sitting position	1.188	0.944	0.764	0.938	0.939

## Data Availability

The data presented in this study are available on request from the corresponding author. The data are not publicly available due to restrictions e.g., privacy and ethical.

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
