# Peer review of "Hole Matrix Mapping Model for Partitioned Sitting Surface Based on Human Body Pressure Distribution Matrix"

_healthcare, 2023, doi:10.3390/healthcare11060895_

Round 1

Reviewer 1 Report (Previous Reviewer 1)

Though the authors revised the manuscript, the way to provide response and rebuttal is not clear. Highlighting revised parts is ineffective for the reviewer to evaluate whether previous concerns are addressed.

After another round of careful review, the reviewer still has critical concerns and comments, please see below,

1, A systematic methodology presentation is missing, what are the methods to guarantee an ideal pressure distribution and design the corresponding cushion? A figure illustration is helpful.

2, Section 1, paragraph 3 is not logically sound, some presentation could be misunderstood, e.g. how to ensure "the forward leaning sitting position was maintained for 11.6 min? in a consistent way?

3, Are the four areas discussed in Section 2.1.4 identical to the four zones in Fig. 6?

4, As stated in Section 3.2, how was the ergonomics combined with pressure distribution matrix in Fig. 5 to produce Fig. 6?

5, When FCM was used to cluster pressure points, how many clusters were generated and why they were selected?

6, How many data samples were collected for each subject in the experiment, and please provide the sample rate.

7, To compare with ordinary sponge cushion, how to determine data samples? Does Fig. 14 indicate only ten samples used for comparison? Label the x-axis in Fig. 14, and also please check the details of other figures.

Author Response

We appreciate your professional comments on our articles. As you were concerned there were several issues that needed to be addressed. Based on your suggestions, we have made extensive corrections to our previous manuscript, please see the attachment.

Reviewer 2 Report (New Reviewer)

The study experimentally obtained the ideal pressure distribution model of buttock and thigh support for office workers in forward-leaning and upright sitting postures, reproduce the support provided by mesh materials with elastic materials, and proposed an effective seat design scheme to improve the comfort of office workers.

1. Introduction. P2, l56-58. “The forward leaning sitting position was maintained for 11.6 min, and the upright sitting position was maintained for only 2.8 min, after which the trunk would alternate between forward and backward leaning”. How did the authors get this conclusion? How the time period maintained for these two postures was obtained?

2. Method. Both male and female subjects were recruited for the tests and four males and three females. Usually at least eight subjects of one gender should be included for subject tests.

3. Seven office chairs were tested and they were all mesh fabrics. Were the textile materials of the mesh fabric the same?

4. Figure 1. From the figure 1, it can be seen that the subjects only wore briefs for the sEMG monitoring, but in real office work people will wear trousers outside. The fabric of the trousers has contact with skin and may influence the test results.

5. Results and analysis. Table 3. The unit of the values (pressure) in the table should be added, So as table 4 and table 5. How did the comfort scores in these tables were obtained?

6. How did the ideal pressure distribution was selected? According to the subjective evaluation of the subjects the ideal pressure was selected? Why not according to the tested values of the pressure but according to the subjective evaluation?

Author Response

We appreciate your professional comments on our articles. As you were concerned there were several issues that needed to be addressed. Based on your suggestions, we have made extensive corrections to our previous manuscript, please see the attachment.

Reviewer 3 Report (New Reviewer)

1. Introduction can be modified, more recent studies related to all the 3 objectives can be included.

2. Results should be discussed with some more recent and related studies.

Author Response

We appreciate your professional comments on our articles. As you were concerned there were several issues that needed to be addressed. Based on your suggestions, we have made extensive corrections to our previous manuscript, please see the attachment.

Reviewer 4 Report (New Reviewer)

The number of participants and possible postures can be increased.

The awkard postures that occur during the use of other possible components (phone, computer, etc.) in the offices can be taken into account.

Author Response

We appreciate your professional comments on our articles. As you were concerned there were several issues that needed to be addressed. Based on your suggestions, we have made extensive corrections to our previous manuscript, please see the attachment.

Round 2

Reviewer 1 Report (Previous Reviewer 1)

With a more rigorous experiment design, the research soundness can be improved in future works.

Reviewer 2 Report (New Reviewer)

It is much better after revision and can be accepted. 

This manuscript is a resubmission of an earlier submission. The following is a list of the peer review reports and author responses from that submission.

Round 1

Reviewer 1 Report

This paper experimentally investigated the ideal pressure distribution of two sitting postures and designed a prototype cushion. However, this work has some fundamental issues.

 (1)    Since the way to obtain ideal pressure distribution is data-driven, the experimental data significantly affects the result reliability. Only seven students were involved in the experiment, moreover, their morphological parameters are relatively consistent (male 173.1 ± 3.3 cm and female: 162.1 ± 2.2. 5 cm). The morphological characteristic of users greatly influences the pressure distribution. How can the authors prove that the ideal pressure distribution derived from the limited data is representative?

(2)    Why were the mesh chairs selected to produce ideal pressure distribution? The chair design also greatly affects the sensed pressure distribution. Why not choose other office chairs with memory foam?

(3)    There is no justification for selecting two sitting postures to calculate ideal pressure distribution. If this study focuses on comfortable sitting, the reason for these two postures should be clarified since there are various common postures, e.g., hunchback, backward-leaning or left-leaning trunk. There are also different legs’ states, e.g., crossed legs. The evidence for selecting these two postures is weak. If the target is to ensure an ergonomic sitting posture, then why consider a forward-leaning posture? A ‘good’ posture with low ergonomic risk is symmetrical, and a forward-leaning posture could lead to the development of low back pain.

(4)    More significantly, to verify the effectiveness of the ideal pressure distribution, the authors leveraged memory foam to build a cushion prototype and compared it with mesh chairs. The memory foam naturally affects the pressure distribution and reduces peak pressure and the change of pressure gradient. Hence, comparing the prototype with the mesh chair is unfair. If the prototype is compared with a memory foam cushion with regular hole distribution, will there be any differences regarding the pressure distribution?

(5)    In Section 2.3.1, based on limited experimental data, it is difficult to conclude that the distance between ischial tuberosities is smaller in males than in the female. Would the results be the same if male subjects with different physical characteristics were recruited? Therefore, based on this doubtful conclusion, it is natural to question the reasonableness for classifying the partitioned pressure distribution by gender (Section 3.1).

(6)    Sitting posture directly influences the pressure distribution. For instance, the pressure maps are different for trunk inclination with 60 degree and with 10 degree. How to ensure the consistency of subjects’ sitting postures when testing different chairs?

(7)    In Section 2.1.3, why select the four indicators to evaluate subjects’ comfort?

(8)    In Section 2.2.2, the pressure maps ranked top 10% were selected to reduce the influence of the differences in the subjects’ scores. This is unreliable since this operation radically does not eliminate individual differences. Other operations, e.g., normalization, can be considered for solving this issue.

(9)    Related works are limited, especially the ones investigating pressure distribution of comfortable sitting.

Reviewer 2 Report

Initial comments - This research study should be listed as a pilot study. The number of subjects is very low. This should be addressed. Either there is no need for more subjects or this data represents preliminary results to determine if further study is warranted. 

- I thought this was a good study and well written. My concerns were with some of the methods. 

Subjects -

  + As mentioned the number of subjects is low. Increase the number of subjects.

  + Increase the diversity of the subjects. Grad students are not necessarily representative of the population of individuals that will use office chairs. Many of us using office chairs are older, overweight, and have underlying orthopedic conditions. Using data associated with a healthy group of low/normal weight young adults is fine for a pilot study, but inappropriate for a full evaluation of this topic. 

Methods 

  + Include more "normal" postures associated with sitting. Often subjects will be "slumped" in a chair. Sitting upright or leaning forward may be ideal, but not practical.

  + Evaluate subjects for a greater length of time. Many of us sit in a chair for long periods of time. As the time increases, our posture likely becomes more relaxed. Determining how pressure points (and posture) change over time (in diverse populations) would improve the quality of the study.

  + In many parts of the study there is no data analysis or no significant differences are found. This is likely an issue of the number of subjects. A follow up to this would be that if there are no significant differences then state that. There is no difference between different configurations of mesh chairs. 

  + The section associated with the prototype seat needs to be further developed or adjusted. You had seven initial seats that were evaluated, but which one was used to evaluate the effectiveness of the prototype? It seems like a discussion of the need of a prototype to improve posture and decrease pressure in a specific are would be appropriate. There needs to be more background information about why creating this new seat is important. I am not seeing the explanation of the importance of the new seat.

Overall Comments -

1) You need more subjects

2) You do not spend a similar amount of time on the different aspects of the study. Much more time is spent on aspect 1 than aspects 2 and 3.

3) The practical significance of the subjective posture scoring in the last sections of the manuscript need to be expanded. What is the practical benefit associated with the prototype?

4) There is very little "discussion". It seems to be infused with the results section. There is no "Discussion" section. 

5) If you want to publish the study with the existing data, mention that this is a pilot study. That helps address the low number of subjects. Also, consider dividing this research into two manuscripts. You spend much more time in the beginning addressing the pressure points on the different mesh seats. 
